# Human drone interaction in delivery of medical supplies: A scoping review of experimental studies

**Franziska Stephan**[1,2]*, **Nicole Reinsperger**[3], **Martin Grünthal**[4], **Denny Paulicke**[1,5], **Patrick Jahn**[1,2]

1 Health Service Research Working Group | Acute Care, Department of Internal Medicine, Faculty of Medicine, University Medicine Halle (Saale), Martin-Luther-University Halle-Wittenberg, Halle (Saale), Germany, 2 Faculty of Medicine, Martin-Luther-University Halle-Wittenberg, Halle (Saale), Translationsregion Für Digitalisierte Gesundheitsversorgung (TDG), Halle (Saale), Germany, 3 Department of Internal Medicine, Health Service Research/Nursing in Hospital, University Hospital Halle (Saale), Halle (Saale), Germany, 4 Pharmacy at the Bauhaus, Dessau-Roßlau, Germany, 5 Akkon University of Human Sciences, Department of Medical Pedagogy, Berlin, Germany

* franziska.stephan@uk-halle.de

**Data Availability Statement:** All relevant data are within the manuscript and its Supporting Information files.

## Abstract

### Background

The COVID-19 pandemic, ageing populations and the increasing shortage of skilled workers pose great challenges for the delivery of supplies for people with and without care needs. The potential of drones, as unmanned air vehicles, in healthcare are huge and are discussed as an effective new way to delivery urgent medicines and medical devices, especially in rural areas. Although the advantages are obvious, perspectives of users are important particularly in the development process. Investigating human drone interaction could potentially increase usefulness and usability. The present study aims to perform a systematic scoping review on experimental studies examining the human drone interaction in deliveries of drugs and defibrillators.

### Methods

Two databases (MEDLINE and CINAHL) and references of identified publications were searched without narrowing the year of publication or language. Studies that investigated the human drone interaction or medical delivery with drones in an experimental manner were included (research articles). All studies that only simulated the delivery process were excluded.

### Results

The search revealed 83 publications with four studies being included. These studies investigated the user experience of drone delivered defibrillators, but no study was identified that investigated the human drone interaction in the delivery of drugs. Three categories of human drone interaction were identified: landing, handover, and communications. Regarding landing and handover, the most important issue was the direct physical contact with the

**Funding:** This research was funded by the Federal Ministry of Education and Research Germany through the "TDG - Translational region of Digital Health Care" project [03COV25E]. PJ receives this funding. The funders had no role in study design, data collection and analysis, decision to publish, or preparation of the manuscript.

**Competing interests:** The authors have declared that no competing interests exist.

drone while regarding communications users need clearer instructions about drone´s direction, sound and look like.

## Discussion

The identified studies used technology-driven approaches by investigating human drone interaction in already existing technologies. Users must become integral part of the whole development process of medical drone services to reduce concerns, and to improve security, usability and usefulness of the system. Human drone interaction should be developed according to the identified categories of human drone interaction by using demand- and technology-driven approaches.

## Introduction

The current COVID-19 pandemic, ageing populations and the increasing shortages in health care workforce pose great challenges for delivering supplies to people with and without care needs—especially in rural areas. Therefore, competent delivery of medical supplies is more necessary than ever before. One scenario to provide people with urgent medical necessities could be a drone-based delivery. Drones are unmanned aircrafts, that are used for various purposes in military, public safety, delivery and increasing medicine [1, 2]. The use of drones is widespread: environment and conservation, agriculture, medicine, construction and industry, commercial shipping, law enforcement and traffic surveillance, and education [3]. Drones in medicine and healthcare are frequently used in public health and disaster relief, telemedicine, and medical transport [3]. Drones in public health are used to gather information about the number of patients in need, detect health hazards, and are useful for epidemiology research. Drones in telemedicine are employed for telesurgery and remote diagnosis as well as treatment of patients by the means of telecommunications technology. Drones in medical transport are used to deliver medication (e.g. vaccines, drugs), blood preservations, organs, defibrillators, and other medical supplies [1–6]. Medical drones are identified as a more affordable alternative to air transport for medical supplies compared with helicopters. In areas with mountains, deserts, forests, with a lack of access to roads or with long-distance travel, or to areas that were affected by major natural disasters [1, 4, 7], the potential of drones in healthcare is considered as high [1]. It is worth noting that drones are poised to revolutionize healthcare in health supply logistics and in out-of-hospital settings, however, drone delivery is still relatively new and unproven.

One review focused on experimental studies in the use of drones for healthcare purposes [8]. Of the nine experimental studies identified by the authors, none was conducted in real-life scenarios following simulation methodologies. Issues that arise in real-life situations regarding health outcomes and external factors as well as people's willingness were overlooked [8]. A recent population survey provides information about the attitude, the willingness of use, expectations, and fears of the German population towards delivery drones and air cabs [9]. Overall, the results showed that the majority of the German population is skeptical about the delivery of consumer goods by drone. However, exceptions to this are emergency situations, e.g., the delivery of medications or for the transportation of injured persons to hospital. The authors suggested a greater consideration of social ideas, doubts, and requirements in the design and further development of drone technology. This requires the involvement of users to improve the effectiveness of the utilization of medical delivery to ensure that the process and technique remain patient-focused [9, 10]. In this vein, ethical challenges are related with humanitarian

innovations, including drones. An international review identified three major trends in ethical considerations [11]. They found issues related to harm (i.e., focus on ensuring physical safety, environmental impacts and benefits), justice (i.e., cost-effectiveness, equitable access and stakeholder responsibility), and respect (i.e., technical aspects of information security, considerations of privacy, active community engagement). These considerations imply a higher involvement of users and stakeholders in future research to improve harm-benefit trade-offs, upholding justice, and respect autonomy. To date, reviews often focus on how drones are used for healthcare and its potentials [11–13]. For example, Hievert et al. [12] identified eight healthcare and health-related applications for drones. Drone delivered medical supplies and treatments was the most common identified application followed by environmental monitoring and using drones to deliver automated defibrillators (AEDs) indicating the importance of drones in improving access to health services. The review found out that all studies on drones are at a pilot stage that have not been implemented or adopted in healthcare settings. The identified studies focused on flight path and payload regulations, but there was limited attention to how patients and communities interact with drones. Rejeb et al. [13] identified three main barriers to humanitarian drones in technological, organizational, and environmental (regulatory) factors, called TOE barriers. Among other things, user acceptance is one organizational barrier to drone adoption in humanitarian logistics. The authors recommend applying technology adoption theories as well as using case studies, interviews, or surveys to understand success and failure factors of using drones within humanitarian organizations.

Hence, analysis of human drone interaction is required for understanding users' needs and requirements in dealing and interaction with drones. The role of users, the involvement of users in drone delivery process and the handover of medical devices or urgently needed medicines are important factors in investigating and improving user acceptance. In the end, it is the user who will have to use the drone in everyday life and get along with it. Non-user-centered technologies will fail to be implemented as long as usefulness and usability are not a focus of research. As long as this is the case, people´s fears and concerns about drones will be the main obstacles to implementation although drones have a great potential in medical care and supply.

Within this healthcare supply context, the present paper is interested in identifying human drone interaction in real life scenarios that have received less attention in the ongoing debate. More specifically, the objective is to assess how human drone interaction in medical transport of medicines and devices in experimental studies is discussed in the academic literature. The scoping review addresses the following question:

How does the human drone interaction take place in the delivery of medication or medical device (defibrillator)?

The current review is especially interested in the role of users (i.e., all persons who are requested to interact with the drone, such as patients, nurses, bystanders, victims, or voluntaries independent of age or disease) in the delivery practice and their experiences, in the procedure of drone delivery, the handover, and the health-related outcomes in healthcare supply (medical supply) and health-related applications (supply with medical devices). In this work, therefore, the central notions are "human drone interaction", "delivery", and "medication", or "medical device". The systematic search followed a two-level process. First, the search involved studies investigating human drone interaction in the delivery of medications. Second, the search involved studies investigating the delivery of medical devices, a scenario in which a person needs urgent help. However, we assume differences in the human drone interaction between the delivery of medications and medical devices. The scenarios are separated because in the medicine delivery scenario the person in need of assistance is the person that interacts with the drone while in the medical device delivery process the bystanders interact with the drone.

Nevertheless, in both scenarios, it was expected that the human drone interaction between the patient, people attending a person or health care professionals and the drone are suitable for the investigation of user's experiences, mainly due to the characteristics of the situations.

## Methods

To identify research gaps and to examine the range and extent of research activity in human drone interaction (user´s role, experience, and acceptance) in medical transport of medicines and devices, we performed a scoping review [14]. Following the systematic literature research (SLR), the present scoping review followed the stages (1) identifying the research question, (2) identifying relevant studies, (3) study selection, (4) charting the data, and (5) collating, summarizing, reporting the results [14]. For improving the rigor, comprehensiveness, and credibility of reporting methods and results of the present scoping review, we followed the PRISMA-ScR Checklist (Preferred Reporting Items for Systematic reviews and Meta-Analyses extension for Scoping Reviews) and JBI (Joanna Briggs Institute) guidelines [15]. The PRISMA-ScR checklist is provided in the additional file S1 Checklist.

The scoping review was included in the ADApp study. The ADApp study was approved by the Ethics Committee of the Martin-Luther-University Halle-Wittenberg (protocol code 2021–069 and date of approval May 6, 2021). All participants gave written informed consent.

**Table 1.  Search terms.**

| | |
|---|---|
| **drone-related terms** | drone* |
| | drone aircraft |
| | unmanned aerial vehicle |
| | UAV |
| | unmanned aerial systems |
| | UAS |
| **health-related terms** | medical application |
| | medical |
| | medicine* |
| | surgical application |
| | medical drone* |
| | vaccines |
| | medical device |
| | home care |
| | professional care |
| **delivery-related terms** | delivery |
| | support |
| | medical transport |
| | medical delivery |
| | delivery of healthcare |
| **user-related terms** | human drone interaction |
| | user centered model |
| | user centered design framework |
| | COVID |
| | SARS-CoV2 |
| | corona |

*truncation

## Search strategy

The systematic search was conducted in the databases MEDLINE via PubMed and CINAHL via EBESCO from 1$^{st}$ to 31$^{th}$ May 2021, using search terms related to drones, health, delivery processes, and users (see Table 1). The search strategies were independently drafted by the two researchers FS and NR and further refined through team discussion. The search was followed by analysis of search terms in titles and abstracts. Initially, both reviews and empirical studies were included in the research. Although the present scoping review aims to assess only empirical studies, reviews were initially included for identifying studies that may have been missed in the database search but were included in reference lists of reviews. Duplicate search results were excluded. Next, sources were selected that discussed drones in the delivery of medicines and medical devices in the abstract. After this first screening, reference lists of those studies that were included were screened. The purpose of this search was to identify studies that may have been missed in the initial search. After reference screening, all reviews and duplicates were excluded from further full-text analyses. The sources that were relevant to the review question indexed two categories: drug delivery as well as defibrillator transportation. The final search strategies for MEDLINE and CINHAL for researcher one and two can be found in additional file S1–S4 Tables. A search for gray literature was not performed because the targeted search for systematic and methodological works had priority in the present study. This decision was made due to the fact that initial desk research revealed a large number of reports on the topic which, however, often a lacked scientific foundation.

## Inclusion and exclusion criteria

In this scoping review, all empirical studies with experimental design (empirical methods) were included which involved human subjects (independent of age or diseases) and drones in the medical delivery of medications and defibrillators. This review aims to assess only experimental studies because they provide evidence to support the usability, acceptance and effectiveness of drones in medical delivery processes. Two authors (FS, NR) searched independently for experimental studies in which health-related outcomes in the human drone interaction and medical delivery with drones were investigated (e.g., delivery time, user experience). We did not restrict the search to group comparisons (e.g., inter- or intraindividual comparisons), language and year of publication on purpose. Studies without drone or medical reference, without empirical background or studies using drones in organ delivery, military, or warfare were excluded from further analyses because we are focusing on human drone interaction in civil drone use. Table 2 summarizes the key criteria for the review process.

**Table 2. Summary of review criteria.**

| Review criteria | Description |
|---|---|
| Publication year | No temporal restriction was applied |
| Publication language | No language restriction was applied |
| Publication type | *Inclusion criteria*: Only journal articles with empirical methods (e.g., interview, focus groups, questionnaires); reviews were included only for reference checking<br>*Exclusion criteria*: articles without empirical design (e.g., reports, websites) |
| Concept | *Inclusion criteria*: human drone interaction (e.g., acceptance, usability, user experience, user-centeredness) in real life scenarios |
| Context | *Inclusion criteria*: supply of medicines or medical devices<br>*Exclusion criteria*: no healthcare context (e.g., military or warfare use) |
| Population | *Inclusion criteria*: human subjects (independent of age, disease, background)<br>*Exclusion criteria*: simulation studies without human subjects |

## Study selection and data extraction

Two researchers (FS, NR) independently performed the searches and screened all studies with regard to the search terms in title and abstract as well as for inclusion criteria in order to exclude studies that were not relevant to the review. To increase consistency among researchers, all researchers screened the same studies. If the relevance of a study was unclear from abstract, the full text was screened. It was set a deadline, after which it was agreed that no more studies would be included in the analyses. Researcher 1 identified 19 publications via MEDLINE and ten via EBESCO. Researcher 2 found two studies via MEDLINE and ten via EBESCO. In the event of disagreement in the selection process, a discussion between these two researchers was carried out until the researchers reached consensus. All studies that met inclusion criteria proceeded to data extraction. Final data extraction involving full-text screening of each included study. Two authors (FS and DP) who independently reviewed the full text and made decision on whether to include or exclude the study based on the aims of this scoping review. Disagreements were resolved by consensus. A data-charting form was jointly developed by the two researchers (see additional file S1 Data). The researchers independently charted the data and discussed the results. The form was continuously updated within an iterative process. The researchers abstracted data of all studies on article characteristics (publication year, country of origin, publication type), case numbers, and contextual factors (study design and research question). For final data extraction, data were abstracted on results (e.g., outcomes of interest), critical features (e.g., drone involvement in the process, drone request, role of users, type of user group (trained or untrained), handover), and participant characteristics, and were then compared with inclusion criteria. Since a scoping review does not conduct quality appraisal, critical appraisal was omitted [14]. The researchers grouped the studies according to inclusion and exclusion criteria. In case a study met the inclusion criteria, results and critical features were summarized in a table (additional file S1 Data). The synthesis of the results followed a narrative summary and thematic analyses, both of which integrate qualitative and quantitative evidence though juxtaposition and synthesis of research findings. The thematic analyses further involves the identification of prominent themes in the studies and the summarizing the findings under different thematic headings [16].

# Results

## Study selection

The research was conducted in May 2021. The deadline after which no more studies would have been included in the analyses was the 31[th] May 2021. In the initial search, 83 results were retrieved: 71 from PubMed and 22 from CINAHL. After removing duplicates, 58 results were included in the initial screening. Titles and abstracts from these 58 studies were assessed. 41 studies were excluded because they either did not have empirical data, a drone or medical reference, or included drones used in military and warfare. The remaining 17 results were then screened for their references according to the research question. Seven reviews, two perspectives, and eight studies were screened according to references. After reference screening, all reviews and perspectives were excluded from further analyses. One study was added with regard to the delivery process of medicine, resulting in four studies, and two studies for the delivery of medical device process, resulting in seven studies. Next, the identified studies were full-text screened. However, all four studies associated with the delivery of medicines were subsequently excluded because they did not involve human subjects or the delivery of organ samples. For the defibrillator delivery process, three studies were excluded because the study did not include empirical data or human subjects. **Fig 1** depicts the contribution of all exclusion

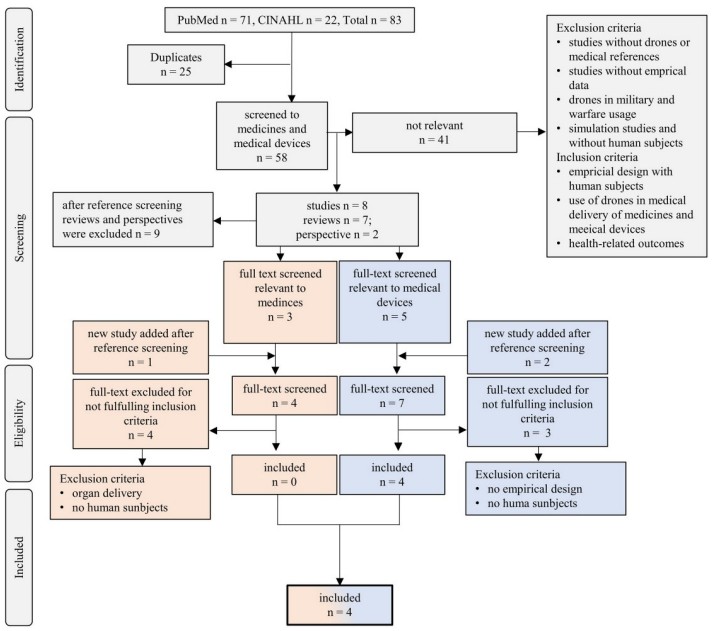

**Fig 1. Frequency of exclusion criteria.**

criteria. Overall, four studies were selected for the systematic synthesis and are summarized in Table 3. See **Fig 2** for the systematic search and study selection process. Please refer to additional file S1 Data for an overview of excluded studies and the reason of exclusion.

## Study characteristics

All eligible studies were written in English. They were published between 2016 and 2020 and were conducted in high-income countries: two in the USA [19, 20] and two in Sweden [17, 18]. However, the data of Rosamond et al. [19] originated in the study of Zègre-Hemsey et al. [20], thus, the study design is similar. Rosamond et al. [19] conducted additional analyses of these data.

**Use of drones.** All four studies addressed an out-of-hospital cardiac arrest (OHCA) [17–20]. The drones were used in a simulated OHCA situation with a manikin where the drone delivers a defibrillator to the site where it is required. Three studies used drones from DJI [18–20] and one study from HEIGHT TECH GmbH & Co. KG company [17]. All studies were funded from Institutes or Foundations that were not related to the manufacturers of the drone. The funders in all studies had no role in study design, data collection, analyses, interpretation of data or producing manuscripts.

**Participant characteristics.** The age range of the participants in the selected studies reached from 18 to 80 years. Claesson et al. [17] did not describe details about involved subjects. However, they described the number of release-methods via parachute ($n = 1$), latch ($n = 6$), or landing ($n = 6$) the drone. It is unclear whether these release-scenarios were tested with 13 different subjects or only with one subject. Two studies involved medically untrained participants who interacted with the drone [18–20]. In one study the participants aged from 73 to 80 years ($n = 8$) [18]. The participants were recruited from a senior citizen organization since persons who suffer from OHCA in Sweden have a median age of 71 years. None of the participants had received a cardiopulmonary resuscitation (CPR) in the last 20 years. In the second study that involved medical untrained participants, the age ranged from 18 to 65 years

**Table 3. Characteristics of the selected studies.**

| Reference | Country | Case Numbers | Study Objectives | Study Design | Role of Users | Health-related Outcomes assessed | Results |
|---|---|---|---|---|---|---|---|
| [17] | Sweden | parachute-release n = 1; latch-release n = 6; landing n = 6 | Benefit of drone system in response time in OHCA; user experience: practical use of drone for delivering AED | Analyzing suitable drone placement using GIS-models and delivery test-flights | User experience in unloading scenarios of defibrillator from bystander | *Delivery time* (models), *AED release scenarios, AED functionality* | UAV were predicted to arrive before emergency in 34% of cases in urban and 93% in rural areas with a mean amount of time saved o 19 minutes; best delivery method: latch-release and landing; parachute-release caused uncertainty about where AED would land; latch-release: no damage and no hurt from rotors; AED should be placed on top of drone for easier access when drone has to land; landing conditions better on flat ground; drone should have sensors, lights, and sounds to attract attention when landing; AED was fully functional |
| [18] | Sweden | 8 (4 females; age: 73–80; mean age 75.5; only woman used smartphones before) | User (bystanders) experience of drone delivered AED in a simulated OHCA-situation; impact of one or two bystanders onsite | Participants were presented to a simulated OHCA situation with manikin; 2 groups: alone or in pairs; instruction that drone would deliver defibrillator, and to call emergency number for help and then follow instruction from dispatcher | Interviews with participants; observations during drone delivery | *Qualitative data*: bystanders experience during drone delivery of defibrillator (content analyses of interviews and observations) *Quantitative data*: time differences between single bystander and dual bystander | *Qualitative data*: 1) technique and preparedness: use of mobile phones most difficult technical moment for participants; difficulties in handling defibrillator despite dispatcher support; participants who expressed a more positive attitude towards technique performed better with regard to all the tasks that they were given; no one hesitated or misinterpreted instruction from dispatcher to retrieve defibrillator delivered from drone 2) support through conversation: interacting with dispatcher gave participants a sense of security, made them confident about retrieving defibrillator via drone (feeling not so scary); long sentences affect situation in negative way 3) aid and decision-making: participants expressed concerns about finding AED fast enough and having direct physical contact with drone; positive expression about the red bag of defibrillator for finding location; participants wish that drone have headlights in order to mark location; participants in alone situation were afraid of leaving person alone; live video streaming from the drone in real time to dispatcher facilitated to assess the situation *quantitative data*: bystanders' hand-off time greater in single situation than in dual situation; delivery via drone useful in situation with more than one bystander; drone delivery increases chance of early defibrillation |
| [19] | USA | 35 (age: 18–65) | User (bystander) experiences of drone delivered AED in a simulated OHCA-situation | AED was delivered by an autonomously flying drone and one bystander searched for a fixed-location AED from the surrounding area (for comparing) | Interviews with participants about their experience with interacting with the drone | *Delivery-time* *Pre-survey*: focus on participant's previous experience with drones, confidence and comfort level with drone interaction, knowledge and confidence about defibrillator and their location on campus, and safety concerns with drones *Post-survey*: participants' impressions about either interacting with drone or experience searching defibrillator | Delivery time faster with drone than searching for fixed AED (ranging from -2 minutes 56 seconds to 1 minute 42 seconds) *pre-survey*: 34% reported that they had previously interacted with a drone *post-survey*: after interacting with drone, 89% reported feeling comfortable during drone approach; 72% had no safety concerns during approach and landing, 85% found it easy to remove defibrillator from the drone; nearly half of participants who were looking for fixed defibrillator reported difficulty in finding it; participants favored use of drone-delivered defibrillator over searching for a fixed defibrillator to not leaving victim alone; concerns about removing defibrillator from drone, fear of landing too close, and uncertainty of drones approaching direction |

*(Continued)*

**Table 3.** (Continued)

| Reference | Country | Case Numbers | Study Objectives | Study Design | Role of Users | Health-related Outcomes assessed | Results |
|---|---|---|---|---|---|---|---|
| [20] | USA | 35 (age: 18–65; ability to jog for 2 min.; no history of cardiovascular disease; paired in sex and age: 18–34, 35–49, 50–65; one single; semi-structured interview with 17 participants) | User (bystander) experiences of drone delivered AED in a simulated OHCA-situation | Participants were presented with simulated OHCA with manikin; stand in pairs; instruction from dispatcher to initiate CPR and instruct second participant to search for fixed defibrillator at one of five zones on campus while autonomous flight of drone that delivered defibrillator was initiated from dispatcher; 17 participants randomized to interact with drone (the other 17 participants looked for fixed defibrillator) | Interviews with participants about their experience with interacting with the drone | *Semi-structured interviews*: focus on experience interacting with drone, concerns, and suggestions for improving design and defibrillator delivery via drone | *Qualitative data*:<br>1) general feelings:<br>generally positive feedback (*exciting*), some neutral, indifferent and negative feelings (uncertainty, anxiety about timing or direction of drone's arrival or landing, landing close enough or too close)<br>2) perceived benefits of drone delivery system: efficient way to get defibrillator, ability to deliver it in remote areas, stay with victim<br>3) concerns and suggestions to improve user experience:<br>a) unclear about drone's location of arrival; drone could emit a noise or something that alerts people of its arrival; reflective tapes or lights to stand out drone<br>b) difficulties to get defibrillator; brightly colored Velcro straps and additional labels useful (e.g., arrow or 'pull here')<br>c) concerns about drone's ability to land effectively or propeller start up again while reaching defibrillator; propellers that tilt upwards or fold and retract after landing useful<br>d) participants needed clearer instructions from dispatcher about what drone would look and sound like, what direction it would be coming from, where it would land<br>e) concerns about defibrillation<br>4) potential use of defibrillator drones in real-lie situations:<br>most would use it in real life; some expressed concerns about drone arrival time, landing in crowed areas, issues with bystanders, flight at night or in poor weather; worries about drone's ability to find location |

*Note*. AED = automated external defibrillator; OHCA = out-of-hospital cardiac arrest; CPR = cardiopulmonary resuscitation

[19, 20]. The participants reported that they have the ability to jog for two minutes and have no history of cardiovascular disease or other medical disease. The participants were paired by gender and age: 18–34, 35–49, 50–65; 18 female pairs and 17 male pairs. One participant performed both roles as a seeker and a caller.

## Human drone interaction in AED delivery

Four studies examined the user experience during the delivery of AEDs [17–20]. In one of the four studies, the involvement of participants is not described in detail, however, three different release-techniques were tested and discussed [17].

In three studies, participants were presented with a simulated OHCA situation using a manikin either indoors [18] or outdoors [19, 20]. In these three studies, participants were instructed to call local emergency number. After the calling, participants had to follow the instructions of the dispatcher to initiate CPR while an autonomous flight of the drone was initiated from dispatcher [18–20]. In one study participants were split into two groups: one group of participants interacted in the situation alone and the other group of participants

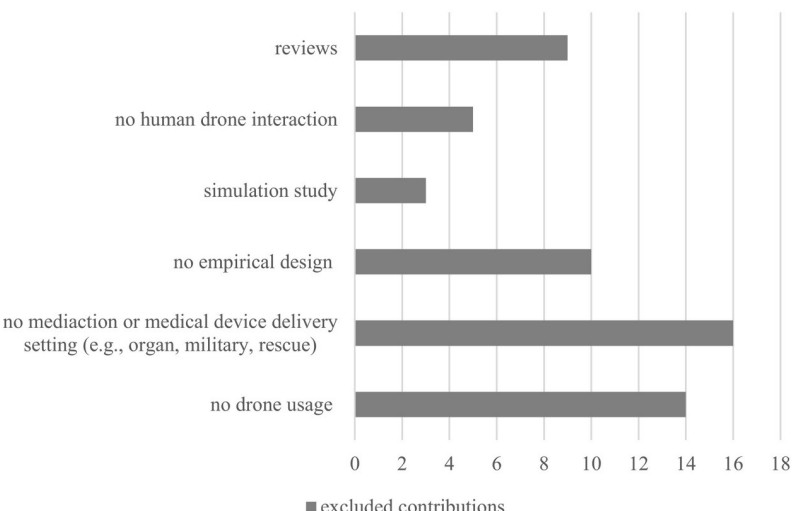

**Fig 2. Systematic search flow diagram.**

interacted in pairs [18]. The other two studies also used a dual situation, but one of the participants was instructed to search for a fixed defibrillator at one of five zones on campus. The other participant was instructed to initiate CPR and to wait for the AED to be delivered by drone [19, 20]. A pre- and post-survey found that 34% of the participants had previously interacted with a drone. Results of the post-survey indicated that 89% of the participants felt comfortable during drone approach [19]. Generally, the participants gave positive feedback about interacting with the drone [20]. Participants who expressed a more positive attitude towards technique performed better on all tasks they were assigned [18]. However, some participants expressed neutral or indifferent feelings such as uncertainty or anxiety. Nevertheless, most of the participants experienced the AED delivered by drone as an innovative and efficient way of obtaining medical help. They noted advantages of being able to deliver AEDs in remote areas and to stay close to the patient in need.

The human drone interaction can be clustered into the following categories: (1) process/landing, (2) handover, and (3) communications.

1. In the four studies identified, the drone landed after arriving at the OHCA location [17–20]. Claesson et al. [17] tested two other arriving scenarios: dropping the AED by using a parachute technique from minimum 25m altitude or dropping the AED using a latch-release from 3-4m altitude. The results showed that the best methods were landing or latch-release. When using a parachute-release technique, wind caused uncertainty about where the defibrillator would land. When using a latch-release, bystanders could fetch the defibrillator. This method of release was suggested as a low risk because people could not be hurt from rotors of the drone. Landing serves as a good alternative, in order to reduce risk for bystanders who want to intervene. However, users had many concerns about the process/landing. Most of the concerns (results of three studies) were related to having direct physical contact with the drone [18–20], although Rosamond et al. [19] found that 72% of participants had no concerns during the delivery process and landing. The users expressed uncertainty that propellers could start when people approached the drone or that the drone would land too close. Consequently, Sanfridsson et al. [18] proposed AED delivery by using a winch that prohibited direct contact with the drone. Participants also suggested propellers that tilt upwards or fold and retract when landing [20]. In this vein, participants also

expressed uncertainty about arrival time of the drone, mainly at night or poor weather [20]. Thus, a drone should emit noise to alert its arrival or have warning sensors to attract attention as well as reflective tapes or lights that would mark its location of arrival [17, 20]. Moreover, participants expressed concerns about the drone's ability to find the right location. Nevertheless, participants favored using a drone which delivers a defibrillator over searching for AED at a fixed station [19].

2. Two studies described the handover process more in detail [18, 20]. In one study participants were informed about at what distance the drone would land from OHCA location (50m). After landing, the drone released the AED and then hovered at 10m altitude to mark the place of AED and to provide visual feedback to dispatcher via live video stream [18]. Only one study described participant's physical contact with the drone in more detail [20]. The AED was fixed with Velcro tapes at the bottom of the drone that had to be removed by the participants. However, one statement that occurred in both studies was the concern about finding the AED fast enough [18, 20]. The hovering of the drone and the red AED bag made it easier for participants to find the right location [18]. All participants felt comfortable with the red AED bag, which made it easier for the participants to find the defibrillator on the ground. In accordance to the process/landing category, participants suggested equipping the drone with headlights in order to mark the location to make it easier to find [18, 20]. The live video streaming from the drone facilitated dispatcher orientation to get an overview about the situation. After landing, two problems that occurred were the difficulty to remove the AED from drone and afterwards handling with the AED [20], nevertheless, 85% of the participants found it easy to remove the AED from the drone [19]. Participants proposed colored Velcro tapes and additional labels that give information about how to remove the AED. Moreover, Claesson et al. [17] proposed a more intuitive location for easier access to the AED, on the top of the drone.

3. Regarding communication, two studies found that participants had uncertainty about the direction from which the drone would come from [19, 20]. The participants needed clearer instructions from dispatcher about the appearance and sounds of the drone as well as the direction it would be coming from, and where the drone would land [20]. However, interaction with the dispatcher was identified as an important source for supporting participants which gave them a sense of security [18]. One problem about communication was that long sentences affect the situation in a negative way. Participants stopped compression for listening to the dispatcher while short sentences affect compression in a positive manner. Thus, Sanfridsson et al. [18] propose the use of short encouraging sentences.

Taken together, the majority of concerns were the physical contact with the drone (process/landing), finding the AED fast enough (handover) as well as the uncertainty about direction and timing of the drone´s arrival/landing (process/landing and communication) [18–20]. The latter was also a result of unclear communication interactions. The most identified user suggestion was to equip the drone with headlights, reflective tapes, or sound to attract attention when drone is coming (process/landing) and to mark its location for release (handover) [17, 18, 20]. Another important suggestion was the need of clearer instructions and short sentences in the interaction between the user and the dispatcher [18, 20]. Table 4 summarizes the user experience and user feedback according to the three human drone interaction categories and its practical implications.

**Health-related outcomes of interest.** As described before, the four studies were interested in the user experience of drone delivered automated external defibrillator (AED) [17–20]. Moreover, two studies intended to explore the delivery time of drone delivered AED

**Table 4. User experience, feedback, and practical implications.**

| | problems | users´ concerns | users´ perceived support | suggestions of users | advantages from the researchers´ point of view | practical implications |
|---|---|---|---|---|---|---|
| **(1) process / landing** | | uncertainty of arrival time [19, 20] | | noise emitting or something that alerts drone arrival [20] | | test flights at day and night, using headlights, warning sensors, reflective tapes or sound |
| | | flight at night [20] | | reflective tapes or lights for marking location of arriving [20] | in case of landing: warning sensors, lights, and sound to attract attention [17] | |
| | | direct physical contact with drone: drone too close or propeller starts when approaching drone [18–20] | | upward inclined or retractable propeller [18] | use a winch to avoid contact with drone [18] | release with winch over landing |
| | | landing in crowd areas [20] | | | | |
| | | issues with bystanders [20] | | | | |
| | | poor weather [20] | | | | investigating complementary and interoperability of drones with other technologies |
| | | drone´s ability to find right location [20] | | | | using applications with GPS tracking |
| | using mobile phones [18] | | | | | educational training |
| **(2) handover** | | finding AED fast enough [18, 20] | drone´s hovering to find right location [18] | headlights in order to mark location of release [18] | live video streaming for facilitation of dispatcher orientation [18] | documentation of handover via live video streaming or app-based confirmation |
| | | | red AED bag for easier identification on ground [18, 20] | | | |
| | difficulty in removing AED from drone [20] and difficulties in handling AED despite dispatcher support [18] | | | colored Velcro tapes and additional labels with removing instructions [20] | suggestion of more intuitive location, e.g., on the top of drone [17] | using applications with instructions, e.g. via video |
| **(3) communication** | | anxiety / uncertainty about direction or timing of drone´s arrival/landing [19, 20] | interaction with dispatcher gave sense of security [18] | clearer instructions from dispatcher about appearance, look and sound of drone, direction, and landing [20] | | clear and short instructions, education of professionals (e.g., dispatcher, pharmacists) |
| | long sentences affect situation in negative way [18] | | | | short encouraging sentences observed positive effect on compressions [18] | |

compared to the time a bystander needed for searching for a fixed defibrillator [19] or to emergency medical services (EMS) [17]. Their results showed that the delivery time decreases when the defibrillator was transported via drone compared to ground search method (ranging from -2 minutes 56 seconds to 1 minute 42 seconds). The difference in response time for EMS was higher for rural than for urban areas (mean amount of time saved of 19 minutes). Another study was interested in time differences between a single bystander and dual bystander in a OHCA situation [18]. Their results showed that the bystander's hand-off time was greater in the single situation compared to the dual situation. Claesson et al. [17] were additionally interested in the functionality of the AED after different release-techniques. The results showed

that the AED was fully functional after its release, independent of the release-method. One study examined the best suitable placement of drones in Stockholm County by using a geographic information system (GIS) tool [17]. To find the suitable places for drones, the authors modeled the EMS delays and OHCA incidences and found twenty suitable locations in urban areas and ten in rural areas.

## Discussion

To date, reviews focus on the applications, benefit, and challenges of drones in society [3, 11–13]. In addition to the increased use of drones in environment, conservation, and agriculture, drones are getting more and more attention in healthcare [3]. In order to realize the use of drones in the sector mentioned, thought must be given to how human drone interaction can be designed. To date, the role of users received little attention. Analyzing the literature focusing on human drone interaction and user experience in the context of medical supply of medicines and devices can provide insights on recent developments and future steps. Thus, the present scoping review is the first reviewing the current state of the literature on the human drone interaction in healthcare, especially in the delivery of medical goods such as devices and medications. This perspective is essential to develop comparative understanding how human drone interaction should be designed with regard to the requirements within the healthcare supply process.

Although Hiebert et al. [12] identified drone-based delivery of medicines as the most common identified application in healthcare, the present scoping review could not identify a study investigating the human drone interaction in the delivery of medicines with empirical design. While a lot of media report about projects using drones in the delivery of medications [21–26], blood [27–29], or other medical goods [30], no empirical study was conducted and therefore, not evaluable for the present scoping review. However, these projects show the variety of scenarios and possibilities for drone use. There is little scientific evidence about the effectiveness, user experience and acceptance of medical drone delivery. Nevertheless, projects in different countries, e.g. in Ghana [21], Tanzania [22, 23], Vanuatu [24], or Germany [25], show that medicine transportation can be implemented. In Ghana drones deliver birth control, condoms, and other medical supplies from a warehouse in an urban area to rural areas where local health workers pick up the supplies [30]. In Rwanda, drones deliver blood to clinics located in remote areas. Health workers request units of blood products via text message and 30 minutes later the blood parcels were dropped on parachutes via drones [27]. Another project in Malawi shows that drones can transport dried blood samples for early infant diagnosis of HIV [28, 29]. Drones can also deliver essential medicines like anti-venom or vaccines to hospitals from one building to another or to remote areas. In China, medical samples and quarantine materials were delivered via drones during the COVID-19 epidemic from hospitals to disease control centers in urban areas. This transportation reduced contact between samples and staff; furthermore, it reduced the time of delivery, providing a more efficient transportation for epidemic prevention and control [26]. One recent study calculated the number of drones and delivery times for the transportation of antiepileptic drugs from an urban hospital to pharmacies, gas stations, and mosques. The authors compared these to delivery times of transportation by road vehicles and found that drones dramatically reduced deliver time for routine and emergency delivery [31]. Nevertheless, the study used simulation methods and thus, human drone interaction was not assessed. These projects imply that the potential of drones in healthcare is enormous. Studies that investigate the human drone interaction by including user experience in the delivery of medicines are still lacking. However, to our understanding the interface of human drone interaction is the key to develop a service that is fitting to patient's and health

care provider's needs and to the speed at which they adopt new technologies. Investigating user´s experience is important because the acceptance of drones will largely depend on how the process is perceived by the users. Thus, involving the users is important so that we know under which aspects drones have added values.

However, the present scoping review could identify human drone interaction in the delivery of medical equipment only. Four eligible studies investigated the impact of drone systems in the delivery of defibrillators [17–20]. Three studies were identified that explored the user experience during the delivery of AED [18–20]. In one study that also examined the delivery of AED via drone, the involvement of participants has not been described. However, the study tested and discussed three different release-techniques [17]. There is consensus that early defibrillation is a key intervention in OHCA situations in that a drone delivery system can compensate for EMS delays whereby the beneficial effect of reduced dispatch times of drones seems higher in rural areas. Thus, drones have a great potential in reducing time to first defibrillation [17]. Participants felt comfortable during drone approach and would use this system in a real-life emergency [17]. This is in line with findings of a survey that people would prefer drones in emergency situations [9]. However, the physical contact with the drone is one of the biggest concerns from users [18–20]. This is an important finding for the technical development of drones. Thus, human drone interaction can be adapted to the user experience, for example by using a winch that reduces direct physical contact with the drone. Another important issue is the communication between the user and the dispatcher. Talking with the dispatcher gave the participants a sense of security [18]. However, users needed more information about the direction the drone comes from, the drone's look and sound, as well as the landing and handover process [20]. Another concern was about the location of landing or handover. Participants preferred lights, reflecting tapes, and sounds that attract attention what is especially useful in emergencies, at night, or at poor weather [17, 18, 20]. Another major issue was to be found in the concern about finding the AED fast enough [18, 20] which was addressed in two studies. Users expressed that a bright color (red AED bag) facilitated the identification of the medical device on the ground [18, 20].

## Practical and theoretical implications

As this scoping review shows, human drone interaction in healthcare is barely the subject of research. Although reviews showed the potentials of drones in the delivery of medicines and medical devices [3, 12] and the barriers to drone adaption in humanitarian logistics, it is surprising that the role of users gets less attention [13]. While there only exist a few studies investigating user experience in the delivery of AED, however, we could not identify a study that investigated the human drone interaction in the delivery of medicines. The scoping review identifies three categories of human drone interaction: process/landing, handover, and communication. Concerns that participants expressed in the studies are useful for designing further drone systems. The three identified human drone interaction categories can be considered in a broader context of user acceptance and need-satisfaction. The technology acceptance model (TAM) shows that better usability and higher usefulness of a technology lead to a higher acceptance of the same [32, 33]. This is important because limited acceptance can be a restricting factor for further drone implementation in healthcare [13, 34]. The TAM describes that the attitude towards using a technology depends on perceived usefulness and ease of use. Perceived usefulness refers to the degree to which a user believes that a technology would improve his or her job performance. Perceived ease of use is defined as the degree to which a user believes that a technology would be free of difficulty and great effort. These determinants are in turn influenced by individual differences, system characteristics, social

influences, and facilitating conditions [35] and can be enhanced through increased experience and knowledge [34]. However, task competence and task autonomy is shown to predict perceived usability implying the impact of psychological factors [36]. From this perspective, three universal and basic psychology needs (autonomy, competences, and relatedness) are assumed to be necessary for optimal human functioning [37]. Based on this, Peters et al. [38] developed the METUX model (motivation, engagement, and thriving user experience) for analyzing the extent of content, functions, and features of technologies in order to improve user experience and wellbeing. In other words, need-satisfaction will increase perceived usability that in turn drives the perceived ease of use, which predicts together with perceived usefulness the actual usage of a technology. According to the results of the present scoping review, the results can be assigned to the three universal and basic psychological factors. Uncertainty and anxiety about the lack of knowledge of the drone´s look, sound, direction, and arrival time was the most identified response of users (measured in terms of the number of studies) [18–20] and can be assigned to the need of autonomy implicating the importance of the need. Autonomy refers to the need of having meaningful choices for behavior. Sufficient information about the drone´s status could make the user feeling agency. Results also show that users felt comfortable with communicating with dispatcher, which gave them a sense of security [18]. This could refer to relatedness, which describes the feeling of being connected to others, feeling cared for by others, and contribute to others. Competence refers to feel able, effective, and mastery in interacting with one´s environment. This need is expressed by user comments about the difficulties removing AED from drone, handling with mobile phones, and the suggestions of additional instructions to help them interact with the technology [18, 20]. Thus, more aspects are requiring attention as potential influences on drone acceptance in healthcare including technology acceptance and need-satisfaction. One way to lead it positively and to further increase acceptance of drones in healthcare settings could be the involvement of users in designing and testing drone-based delivery of medical goods in the sense of "user-centered-designs". Three of the included studies [18–20] considered the user experience in this sense. A user-centered design (UCD) is an evidence-based approach that plays a key role in achieving user engagement [10, 39]. There are two approaches that can be distinguished in the designing of technology: In the technology-driven approach, the developmental process is primarily determined by possibilities of already existing technologies. However, the needs and wishes of users are limited within this framework. In the demand-approach, these needs of users were integrated especially before technologies were developed [40]. However, the three included studies [18–20] used the technology-driven approach. In this vein, a change of perspective seems to be indicated. It seems necessary to involve users in drone systems as early as possible to reduce concerns and negative feelings, and to improve the usability and usefulness through need-satisfaction of the system in real life. This is especially important when the person who needs help is also the person who waited for the drone.

As shown in **Table 4**, the advantage of user involvement is that the technology can be further developed according to needs and requirements, and many implications can be derived for system development. Users can identify strength and weaknesses of the system and can add important information regarding improvements of the system. Regarding process/landing test flights at day and night seems useful to simulate different supply scenarios. Drones should have headlights, warning sensors, reflective tapes or should make sounds to attract attention and to mark location of arriving [17, 18, 20]. To avoid direct physical contact and thus reduce anxiety, operators should prefer release with winch over landing [18, 19]. Users expressed concerns about the uncertainty about the timing, direction of drone´s arrival and drone´s ability to find the location [18–20]. These concerns are important feedbacks regarding process/landing and communication. This implies that users, especially these persons who are in need,

require information about the delivery status and interaction with dispatcher to get a sense of security. The appearance of these concerns in three studies implicate how important the information about drone status are for users. This can be solved via applications that track the drone so that the user is informed. Regarding communication, dispatchers should be trained according to how the information are presented to the user to support him or her [18]. Future studies should investigate which factors contribute to a supporting interaction and conversation between user and dispatcher. Regarding handover, the biggest concern was to find the AED fast enough. Bright colors of the medical goods as well as headlights are important features for supporting users to find the medicines or medical devices fast enough and to reduce negative feelings. Moreover, drone systems should have a documentation of handover, e.g., via video live streaming or an app-based confirmation [18]. Future research should focus on user experience and should investigate users' feedback iteratively.

Within the selected studies, all studies were conducted only in developed industrialized countries such as USA [19, 20] and Sweden [17, 18], although a lot of projects are running in developing countries implying the need of drones in healthcare in such countries. In the future, user´s role and experience should be a focus of research not only in developed countries because needs and requirements might differ between countries with high income as well as developed healthcare supply and countries with low income and low developed healthcare supply. Another point mentioned is the limitation of variation according to drone systems (three studies used drones of DJI) and scenarios. Within the selected studies, a significant focus was laid on the capabilities of drones in the delivery of defibrillators and its impact on health-related outcomes during a relatively narrow OHCA simulation. However, to implement and adopt drones in healthcare settings, a greater approach is needed by investigating the user experience by including all relevant healthcare workers (e.g., persons who communicate with patients/helpers), patients and helpers.

## Limitations

Although the scoping review was supported by steps including refinement of the protocol through team discussion, blinded searching, and selection of articles by two researchers, several limitations have to be mentioned. First, the present study did not include gray literature. Although the search included two relevant academic databases with peer-review journal articles, gray literature might also yield additional insights. Thus, it is possible that studies with human drone interaction focus have been overlooked. Second, the search protocol was based on a combination of keywords that may not capture all relevant studies. Third, the theoretical and practical implications derived from this scoping review need to be tested and validated with empirical research methods (e.g., MRC-Framework). Forth, we adopted a relatively narrow approach to the context by including supply of medicines and medical devices. Expanding the context might contribute to more hits according to human drone interaction settings. Nevertheless, the strength of this narrow context is to provide an insight in specific domains and its potential.

## Conclusion

Given the health-related effectiveness of drones on the reducing of travel time, especially to remote or difficult to access regions [1, 4, 17, 19], future studies may clearly benefit from a more integrated approach to user experience and user engagement in the medical delivery ensuring the drone process remains user-focused. Implementing a new delivery system such as the medical delivery with drones introduces new issues in the interaction with medical or nursing staff, pharmacists, dispatchers, and patients. Thus, at this point, we cannot conclude

how productive or counterproductive the drone system might be in clinical reality. Given the lack of user-centered research in the field of medical delivery, studies are needed that investigate the human drone interaction including demand- and technology-driven approaches. In the next step, it seems functional to investigate the usefulness and usability through mixed-method approaches. For developing technologies that are helpful for both, medical staff and patients, the needs of users should be investigated and translated into functional requirements and design guidelines. Based on this concept, prototypes can be developed for eliciting feedback from users. Thus, users can participate in the technology development. The benefit of an iterative user-centered process is that knowledge comes on the one hand from users, technicians, and the context, and on the other hand from the functioning of prototypes in the real world based on empirical results [39]. However, the studies presented here consider only one group of users: the bystanders in an OHCA situation. To cover the entire process, all user experiences that are important for the medical delivery should be involved in experimental studies including for example medical and nursing staff, pharmacists, or dispatchers.

## Supporting information

**S1 Checklist. Prisma-ScR checklist.**
(DOCX)

**S1 Table. Search strategy for Medline researcher 1.**
(DOCX)

**S2 Table. Search strategy for Medline researcher 2.**
(DOCX)

**S3 Table. Search strategy for CINAHL researcher 1.**
(DOCX)

**S4 Table. Search strategy for CINAHL researcher 2.**
(DOCX)

**S1 Data. Data extraction.**
(XLSX)

## Acknowledgments

We would like to thank the ADApp-Team for exchange and discussion of human drone interaction. Especially, we would like to tank Nicole Reinsperger (NR) for her help with the systematic literature research. Additionally, we would like to thank Jamie Smith for editing works.

## Author Contributions

**Conceptualization:** Franziska Stephan, Patrick Jahn.

**Formal analysis:** Franziska Stephan, Martin Grünthal.

**Funding acquisition:** Patrick Jahn.

**Investigation:** Franziska Stephan, Nicole Reinsperger.

**Methodology:** Franziska Stephan.

**Supervision:** Patrick Jahn.

**Visualization:** Franziska Stephan.

**Writing – original draft:** Franziska Stephan.

**Writing – review & editing:** Denny Paulicke, Patrick Jahn.

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
