## [Decision Letter · Decision Letter 0]

12 Jan 2022

PONE-D-21-23464Human drone interaction in delivery of medical supplies: a scoping review of experimental studiesPLOS ONE

Dear Dr. Stephan,

Thank you for submitting your manuscript to PLOS ONE. After careful consideration, we feel that it has merit but does not fully meet PLOS ONE’s publication criteria as it currently stands. Therefore, we invite you to submit a revised version of the manuscript that addresses the points raised during the review process.

We look forward to receiving your revised manuscript.

Kind regards,

Rafael Santos Santana

Academic Editor

PLOS ONE

https://journals.plos.org/plosone/s/file?id=ba62/PLOSOne_formatting_sample_title_authors_affiliations.pdf”

2. Thank you for stating the following in the funding Section of your manuscript:

“FKZ03 funding code: 03COV25E” We would like to thank the ADApp-Team for exchange and discussion of human drone in-teraction. A special thank goes to Nicole Reinsperger (NR) for their help during systematic literature research. We also would like to thank Jamie Smith for editing the manuscript.

“This research was funded by the Federal Ministry of Education and Research Germany through the “TDG - Translational region of Digital Health Care” project [03COV25E]. PJ receives this funding. The funders had no role in study design, data collection and analysis, decision to publish, or preparation of the manuscript.”

4. "We note that this manuscript is a systematic review or meta-analysis; our author guidelines therefore require that you use PRISMA guidance to help improve reporting quality of this type of study. Please upload copies of the completed PRISMA checklist as Supporting Information with a file name “PRISMA checklist”."

Additional Editor Comments:

The work is consistent with the editorial proposal of the journal, however it needs some important methodological adjustments and information that helps readers to better understand some of the study's issues should be inserted. The reviewers brought notes necessary for the adequacy of the work

Reviewers' comments:

Reviewer's Responses to Questions

**Comments to the Author**

1. Is the manuscript technically sound, and do the data support the conclusions?

Reviewer #1: Partly

Reviewer #2: Partly

2. Has the statistical analysis been performed appropriately and rigorously? 

Reviewer #1: N/A

Reviewer #2: N/A

3. Have the authors made all data underlying the findings in their manuscript fully available?

Reviewer #1: Yes

Reviewer #2: No

4. Is the manuscript presented in an intelligible fashion and written in standard English?

Reviewer #1: Yes

Reviewer #2: Yes

5. Review Comments to the Author

Reviewer #1: The theme is exciting and has an interesting and current approach. I believe that some adjustments are needed to bring more fluidity to the work and the reader's attention. The introduction is well written, but it is worth reviewing/restructuring considering a macroscopic perspective from the use of drones to the key idea of the study ("medical delivery processes"). In the methodology, clarify what the asterisks in table 1 refer to. Clarify what "Source types included reviews" is (line 115) given that "This review aims to assess only experimental studies because they provide evidence to support the use, acceptance and effectiveness of drones in medical delivery processes" (129-131). I suggest better describing "search update" (166) (would it be: "references of identified publications were searched" in lines 28/29 of the abstract?). In the results, the section "Human drone interaction in AED Delivery" could be structured in a more fluid and interesting way for the reader. It is a fundamental (and very interesting) part of the work, but it can be presented in a more objective way and even with the use of comparative tables. In the discussion I believe that there can be a greater approach on: 1. the articles were only developed countries (although, as presented in the article, there is use of drones in several countries); 2. equipment only (all four studies addressed an out-of-hospital cardiac arrest; 3. three studies used drones from DJI), little variation; 4. focus on rural areas. Add limitations that could potentially influence the results obtained.

Reviewer #2: Review comments on “Human drone interaction in delivery of medical supplies: a scoping review of experimental studies”

Thank you for allowing me the opportunity to read your manuscript and provide feedback on your work. I respectfully submit the comments to help improve your article.

1. Summary of the research and overall impression

The work addresses the use of drones with application in public health. This is a relatively new topic and about which, in the coming years, growth in the literature should be observed. The methodological choice of a scoping review is pertinent and adequate, although there is a need to add more information to improve the clarity of the work and allow better compression. I have pointed out some critical issues that I would like the authors to elucidate. I hope the comments can be useful to improve the quality of the work. Thank you again for allowing me to review your work.

2. Major issues

Introduction

2.1. Line 78: in a non-exhaustive search in Pubmed, I found a scoping review that analyzed the use of drones in humanitarian contexts (DOI: 10.1007/s11948-021-00327-4). Authors should review the phrase that refers to the non-existence of scoping review on the subject and analyze the relevance of including the referred work in the review.

2.2. Lines 86-99: authors are requested to include more details about population and context, in line with the mnemonic PCC (population, concept, and context) found in the JBI 2020 manual (https://doi.org/ 10.46658/JBIMES-20-12).

Methods

2.3. Line 103: the authors cite PRISMA 2020 as a method for systematic search. The text should be reviewed since PRISMA is a guideline for reporting systematic reviews and not a guide for their preparation. Due to my language limitation in German, I was unable to evaluate the full text of citation #13 which references a guide for conducting scope reviews, based on the methodology returned by JBI. It should be noted that the official JBI 2020 guidelines for conducting scoping reviews can be accessed at https://doi.org/10.46658/JBIMES-20-12 and should preferably be used in references to the method for preparing a scoping review.

2.4. Line 105: the authors cite the preprint of the manuscript containing the PRISMA update for systematic reviews (DOI: 10.31222/osf.io/v7gm2). However, the article has already been published in several journals (https://www.equator-network.org/reporting-guidelines/prisma/). Considering the existence of the PRISMA Extension for Scoping Reviews (https://doi.org/10.7326/M18-0850), it is requested that this instrument be used instead of PRISMA for systematic reviews.

2.5. Line 146-147: the manuscript reports that the final data extraction was performed by one of the authors. There is no mention of whether a second author reviewed the extraction and whether there was an assessment of consensus among the authors. Please submit more information.

2.6. The authors do not mention whether a search for gray literature was performed. Please justify if it has not been performed.

2.7. The authors did not report if a one priori protocol for the scoping review was created. Scoping review protocols can be registered on platforms such as “Open Science Framework” or “Figshare”, or even published in journals. If it has been created, it is requested that the code and a source be included to access it.

Results

2.8. According to the JBI 2020 manual, authors are requested to present, as supplementary material, a table with excluded works and the reasons for exclusion.

Discussion

2.9. Authors are asked to present a topic/paragraph to address the strengths and limitations of this scoping review.

3. Minor issues

Introduction

3.1. Lines 71-77: citation #9 appears to be missing.

3.2. I suggested that the authors justify/specify details about the originality of the review compared to the reviews already published.

Methods

3.3. Lines 139-154: if the authors have used a template tool for data extraction, please include it in the additional information section.

Discussion

3.4. Lines 345-346: authors must insert reference(s) to support the statement presented in the discussion.

3.5. Lines 348-357: authors must insert reference(s) to support the statement presented in the discussion.

3.6. Line 370: please inform which are the three studies.

3.7. It is suggested that the authors discuss the results further in the context of the previous reviews identified.

References

3.8. Some references are incomplete (eg, 21, 22, 23, 25, 26, 28, 29) and in disagreement with the journal's guidelines.

6. PLOS authors have the option to publish the peer review history of their article (what does this mean?). If published, this will include your full peer review and any attached files.

Reviewer #1: No

Reviewer #2: **Yes: **Evandro de Oliveira Lupatini

---

## [Author Response · Author response to Decision Letter 0]

4 Mar 2022

Dear Reviewers,

we are very thankful for the constructive comments provided by you. In the following, we address all comments in a point-by-point fashion. Changed paragraphs were marked in blue color throughout the manuscript. 

Thank you very much.

Yours sincerely,

Franziska Stephan (corresponding author), on behalf of all co-authors

---

## [Editor Report · Decision Letter 1]

13 Apr 2022

Human drone interaction in delivery of medical supplies: a scoping review of experimental studies

PONE-D-21-23464R1

Dear Dr. Stephan,

We’re pleased to inform you that your manuscript has been judged scientifically suitable for publication and will be formally accepted for publication once it meets all outstanding technical requirements.

Kind regards,

Rafael Santos Santana

Academic Editor

PLOS ONE

Additional Editor Comments (optional):

The authors made the requested changes properly. The work is relevant and will contribute to the scientific knowledge of the area

---

## [Editor Report · Acceptance letter]

19 Apr 2022

PONE-D-21-23464R1 

Human drone interaction in delivery of medical supplies: a scoping review of experimental studies 

Dear Dr. Stephan:

I'm pleased to inform you that your manuscript has been deemed suitable for publication in PLOS ONE. Congratulations! Your manuscript is now with our production department. 

Kind regards, 

on behalf of

Dr. Rafael Santos Santana 

Academic Editor

PLOS ONE